# Body Odours Sampled at Different Body Sites in Infants and Mothers—A Comparison of Olfactory Perception

**DOI:** 10.3390/brainsci11060820

**Published:** 2021-06-21

**Authors:** Katharina Hierl, Ilona Croy, Laura Schäfer

**Affiliations:** 1Department of Psychotherapy and Psychosomatics, Technical University of Dresden, 01062 Dresden, Germany; Laura.Schaefer@uniklinikum-dresden.de; 2Department of Biological and Clinical Psychology, Friedrich-Schiller-University of Jena, 07743 Jena, Germany; Ilona.Croy@uni-jena.de

**Keywords:** body odour, sampling methods, body parts, body sites, family odours, chemosignals, odour sampling, kin recognition

## Abstract

Body odours and their importance for human chemical communication, e.g., in the mother–child relationship, are an increasing focus of recent research. Precise examination of sampling methods considering physiology and feasibility aspects in order to obtain robust and informative odour samples is therefore necessary. Studies comparing body odour sampling at different body sites are still pending. Therefore, we sampled axilla, breast, and head odour from 28 mother–infant dyads and examined whether odour perception differs with regard to the body site. The participating mothers were asked to evaluate their own and their infant’s body odour samples, as well as odours of two unfamiliar mother–infant dyads. We tested whether maternal pleasantness and intensity evaluation, as well as recognition ability of the odours differed between the body sites. In infants, the head odour exhibited slightly lower pleasantness ratings than axilla and breast, and intensity ratings did not differ between body sites. In mothers, body site affected intensity ratings but not pleasantness ratings, as the breast odour was rated as less intense compared with head and axilla. Across all body sites, mothers rated the own and their infant’s odour as less intense when compared with unfamiliar samples. Recognition ability did not differ between body sites, and in line with previous studies, mothers were able to recognize their own and their own infant’s odour above chance. In sum, our study extends the previous methodological repertoire of body odour sampling and indicates that the axilla, breast, and head of adults as well as infants serve as informative odour sources.

## 1. Introduction

Human communication is strongly informed by body odour (BO) [1,2]. Odours provide information on the emotional state of the sender [3], mediate attractiveness perception [4], signal familiarity [5], and are a source of comfort [6]. Those features are of particular relevance in intimate relationships, such as between parents and children or between romantic mates. However, the investigation of BO poses methodological challenges. A comprehensive study preparation requires careful consideration of standardisation protocols for the participants, the sampling medium and duration, and how the samples should be presented experimentally. The selection of a suitable body site with regard to sweat physiology and sampling feasibility according to the developmental stages of the participants needs to be considered in particular. Experimental investigations on that regard are however still pending.

In the following, a short overview of previous methodological assessment on body odour sampling is presented (see Table 1). Standard odour sampling protocols aim at controlling various external and internal sources of odour contamination to keep the odour as clean as possible. First, the influence of perfume and fragranced hygiene products which are widely used in Western 21st century civilisation must be minimized. Odour donors are therefore asked to refrain from using deodorants, antiperspirants, or other perfumed hygiene products and are only allowed to use odourless products, such as specific soap (e.g., [7,8,9]). Second, many studies set a range of eating rules, as some foods change perceived BO quality. For example, Havlicek and Lenochova [10] found that the consumption of red meat negatively affects BO quality. Conversely, participants of Zuniga and colleagues [11] report that meat, in addition to tofu, eggs, and fat, would induce a pleasant BO. Although research has been inconsistent on that topic, a number of specific instructions are frequently applied, which include avoidance of garlic, blue cheese, and alcohol [7,9,12,13]. Third, odour donors are instructed not to smoke during the sampling period. Fourth, odour sampling has to be conducted with garments being pre-washed with odourless detergent, in order to prevent contamination with perfumed detergents or fabric softener. In addition, clothes being worn together with the sampling garment have to be prepared in the same way [8,14].

The next question to address is which medium to use for sampling. Yao and colleagues [15] investigated the absorbance capability of volatile compounds of BO in different fabrics. They found that polyester absorbed more odour compounds than cotton and wool. Similarly, Callewaert et al. [16] reported that BOs that had been sampled with polyester smelled more intense and less pleasant, which could be linked with dissimilar bacterial growth when compared to cotton. Nonetheless, according to this study, the microbial profile of polyester clothing rather leads to malodour [16]. In the previous literature, the most common sampling mediums are thus either cotton pads [13,17,18,19] or cotton t-shirts [8,12]. While t-shirts are simply worn by participants, cotton pads are usually fixed directly to their skin with surgical tape. This impairs feasibility when sampling infants’ BOs, as the tape can lead to itching or removal of the tape, as well as to skin irritation during removal.

Further, the duration of BO sampling should be carefully considered, as it affects perceived BO quality. According to Havlicek et al. [20], shorter worn cotton pads (12 h) are perceived as more pleasant, more attractive, and less intense than longer worn cotton pads (24 h). The sampling durations to be found in literature vary from 30 min [21] up to 7 consecutive nights [22]. An additional influencing factor for assessing BOs in women is the menstrual cycle. The female BO leads to a more pleasant odour imprint in the fertility phase of the cycle, irrespective of whether men [23] or women [24] rate the BO. In line with these findings, Lobmaier et al. [19] have shown that a higher level of oestradiol and a lower level of progesterone relate to the attractiveness ratings of female BOs.

The experimental presentation of the odour samples involves further methodological difficulties. Using freshly sampled BOs (e.g., [25]) involves organizational and logistical challenges to coordinate sampling and assessments accordingly. Therefore, it is more feasible to store the BO samples until experimental presentation. As Lenochova et al. [7] investigated, BOs are conservable in a frozen state (at least −32 °C) for up to 6 months without substantial reduction of BO quality. After this time period, perceived intensity of the thawed BO samples decreases, while other odour qualities are not affected.

Finally, the addressed body sites must be chosen. 

In view of human sweat physiology, the stage of development of the odour donor and, in particular, secretion of glands must be considered. In adults, the axilla is found as body site of choice when assessing odour-related perception with experimental paradigms [7,13,17,26].

**Table 1 brainsci-11-00820-t001:** Overview of methodological approaches to body odour sampling.

Study	N	Design	Assessed Factor	Medium and Body Site	Notes	Results
Fialová et al. 2019 [17]	*n* = 12 female odour donors, *n* = 56 male odour raters	within subject	Caloric intake, BO samples 3x from each donor (during habitual diet, during 48 h of caloric restriction, 72 h after caloric restauration	Cotton pads, axilla	Examination of general health status (blood and urine tests), dietary restrictions	Hedonic ratings: odour samples from restoration phase were significantly more pleasant, attractive, less intense compared with previous phases
Lenochova et al. 2009 [7]	*n* = 9 male odour donors, *n* = 14 to 21 female odour raters	within subject	Exp. 1: frozen storage (immediately after sampling, after 2 weeks, 4 weeks, 16 weeks) and repeated thawing (1, 2, and 3 times) of BO samples; Exp. 2: frozen storage for 6 months	Cotton pads, axilla	Dietary restrictions	Exp. 1: frozen storage: effect on intensity (higher intensity after 4 weeks of storage compared with fresh and 16 weeks of storage); repeated thawing: lower intensity in repeatedly thawed samples; for both storage and thawing no effect on hedonic qualities
	*n* = 25 male BO donors, *n* = 27 female odour raters	within subject	Exp. 2: frozen storage of BO samples for 6 months without thawing	Cotton pads, axilla	Identical restrictions as in Exp. 1	Exp. 2: no effect on intensity or hedonic qualities (pleasantness, attractiveness, masculinity)
Kohoutová et al. 2012 [13]	*n* = 11 male odour donors, *n* = 30 female odour raters	within subject and control cond.	Shaving of axillary hair (either regularly shaved or unshaved)	Cotton pads, axilla	Dietary restrictions	Effect of shaving: shaved armpits were perceived more pleasant, more attractive, less intense; perceived intensity was increasing with growing back axillary hair
Havlíček and Lenochova 2006 [10]	*n* = 17 male odour donors, *n* = 30 female odour raters	within subject	Meat consumption	Cotton pads, axilla		Samples in the non-meat phase were rated more pleasant, more attractive, and less intense
Havlíček et al. 2011 [20]	*n* = 7 male BO donors, *n* = 25 female BO raters	within subject	Sampling length: 12 h versus 24 h of sampling	Cotton pads, axilla	Dietary restrictions	Shorter worn cotton pads (12 h) were perceived more pleasant, more attractive, and less masculine and intense
Fialová et al. 2016 [27]	*n* = 42 male BO donors, *n* = 82 female BO raters	within subject	Garlic consumption	Cotton pads, axilla		The odours of subjects in the garlic condition were rated as more pleasant, attractive, and less intense
Roberts et al. 2013 [12]	*n* = 92 male stimulus providers (olfactory, visual), *n* = 63 female raters	within subject	Repeatability of odour perception	Cotton t-shirts, whole garment	Dietary restrictions	The perceptions of all stimuli were repeatable
Callewaert et al. 2014 [16]	*n* = 26 odour donors (13 male), *n* = 7 odour assessors (selected and screened)	between subject	Microbial odour profile after 28 h and qualitative ratings for different fabrics (cotton versus polyester)	Subject of investigation, complete T-shirt	T-shirts were stored at room temperature, so that bacteria could grow	Cotton t-shirts smelled more pleasant and less intense than the polyester t-shirt; different growth of bacteria in the fabrics: micrococci occurred almost exclusively in polyester, staphylococci occurred in both fabrics, and corynebacterial in neither of the two
Zuniga et al. 2017 [12]	*n* = 43 male odour donors, *n* = 9 female raters	between subject	Diet quality (fruit and vegetable intake, and skin yellowness)	Cotton t-shirt	Diet quality was measured (i) directly with self-report questionnaire on dietary habits and (ii) indirectly using skin yellowness	Skin yellowness correlated significantly with positive affective odour ratings, no relation of yellowness and BO intensity. Self-reports: consumption of eggs, tofu, oils, and fats was associated with positive ratings; seafood and carbohydrate consumption was associated with more negative and more intense ratings
Gildersleeve et al. 2012 [23]	*n* = 41 female odour donors, *n* = 112 male odour raters	within subject	Menstrual cycle (high fertility versus low fertility state)	Cotton pads	Fertility state was validated with ovulation tests	Men were able to reliably discriminate between high- and low-fertility samples; high-fertility odours were preferred; discrimination ability was associated with even more pleasant evaluation of the high-fertility samples

Sergeant [28] describes the axilla as a unique source of BO due to the range of different secretions and the highest density of glands. Additionally, the axillary odour seems to be particularly relevant for communication via pheromones and is a prominent odour source in humans [29]. In addition to the axilla, research hints at another relevant body site in adult women: Varendi and Porter [30] demonstrated that newborns strongly orient and move towards the maternal breast odour compared with a neutral odour. Similarly, Doucet et al. [31] found that maternal breast odour links to infant behaviour as infants were more likely to open their eyes and cried less when they could smell the breast odour than when they were exposed to a covered and thus odourless breast. Apart from the infants’ odour perception, this suggests that the maternal breast provides a characteristic body odour, which emerges from the Montgomery glands. Those glands are distributed on the areolae and enlarge during pregnancy and lactation and are believed to play a key-role in initiating the milk transfer processes [32].

When it comes to investigation of body odours in children, it is important to consider the different secretions of glands emerging during development. While eccrine and apocrine sweat glands already exist at birth, only eccrine glands are already active at birth. Apocrine and eccrine sweat glands become active during puberty and are distributed over genital, perineal, and axillary areas [33]. These glands secrete into the canals of hair and thus become active, parallel to emergence of pubic hair during puberty [34]. Barzantny and colleagues [35] describe that the development of strong BO is linked with secretion of apocrine glands rather than of eccrine glands. These findings lead to the conclusion that the sampling of BOs from axillary, perineal, and genital areas of the body from (post-) pubertal children who already passed this stage provides olfactory impressions with comparable intensity to adult BO. 

To date, however, it how this applies to prepubertal odours has never been investigated. Examining the BO of prepubertal children, a number of studies used axillary sampling, as this has been state of the art across BO research and also proven to transport olfactory information [8,36,37]. Moreover, other studies investigating parental perception of children’s odours lack information on sampling methodology and do not specify the exposed body sites at which the odour was sampled [38,39,40].

Typically, two ways of BO assessment in order to capture human BO perception have been used: rating of BO quality and BO recognition. First, BO quality is measured by pleasantness and intensity perception [38,41], which differs according to, e.g., familiarity of an odour [5]. It yet has been unclear how BO quality differs with respect to different body sites. Second, self-recognition of one’s own BO has been shown in adults [21] but has not been compared for different body sites. The recognition of one’s own child by chemosignals is considered olfactory kin recognition and has been studied in view of its behavioural relevance. Among other purposes, this serves as insurance of the survival of the offspring because in this way rearing resources can be invested in a targeted manner [36] and the formation of an affectionate bond is facilitated [42]. In previous literature, it was shown that olfactory kin recognition is a reliable phenomenon in healthy individuals: Mothers are able to identify their children above chance level [8,25,43]. Therefore, a reliable kin recognition performance may indicate the quality of BO sampling. The authors of Kaitz [25] found a link between the capability of kin recognition and the length of exposure to their infant’s BO. A particularly exposed part of an infant’s body is the head, as shown in the questionnaire study by Okamoto et al. [44]: Mothers reported that they actively seek their infants’ BO in everyday rearing, with the head providing the most common source of affective experiences. These results suggest evaluating the perceived BO quality of infant’s heads, its recognizability, but also its methodological feasibility. 

The current study therefore investigates whether different BO sites affect perceived BO quality and BO recognition for both maternal and infantile BOs. The axilla, the breast, and the head were compared as assumed areas of relevant odour sources. The participating subjects rated their own and their child’s odour, as well as unfamiliar female and infant odour samples. We hypothesized that the infantile odours are perceived as being equally intense at all body sites and most pleasant on the head. Regarding the mothers, we expected the highest intensity ratings for the axilla and lowest pleasantness for the axilla. Further, we aimed to replicate that the familiar odour (one’s own and one’s own infant’s odour) is perceived as being less intense and more pleasant than unfamiliar odours [5,45]. In an exploratory manner, we tested the relationship between pleasantness and intensity ratings for each body site. We assumed that (i) the recognition performance of one’s own infant’s odour is highest for the head odour and that recognition performance of one’s own odour is highest at the axilla. Furthermore, we aimed to replicate that (ii) recognition exceeds chance level at all body sites.

## 2. Materials and Methods

The ethics committee of the University of Dresden (Code: EK 104032015) approved the conduction of the study in accordance with the “World Medical Association’s Declaration of Helsinki”. Written, informed consent was received from all participants.

### 2.1. Sample Description

The sample consisted of 28 normosmic mothers (M = 32.43 years, SD = 4.35) participating with their biological child aged up to 12 months (M = 6.12 months, SD = 2.39). Subjects filled in a questionnaire on their sociodemographic status asking for age, sex, immigration status, highest school-leaving qualification, professional qualification, degree, relationship status, and number of children. Afterwards, they completed information on their medical and psychological status, including the following: disease status, alcohol and smoking habits, exposure to gas or chemicals, pregnancy and birth complications, preterm birth, and serious disease of their baby (see Appendix A
Table A1, Table A2 and Table A3).

### 2.2. Inclusion and Exclusion Criteria

For study inclusion, biological motherhood of at least one child aged up to 12 months was required. Pregnancy, anosmia, and hyposmia were exclusion criteria. Olfactory performance was tested with the screening version of the standardized Sniffin’ Sticks Sep II identification test. This screening test assesses the correct identification of three odours (cinnamon, banana, fish odour) [46], and all subjects were able to correctly identify the presented odours. 

As depression can affect olfactory performance [47], the participating mothers filled in the PHQ-4 (Patient Health Questionnaire, [48]) questionnaire in order to screen for their mental health. Results revealed no impairment due to depressive or anxiety symptoms (M = 1.04, SD = 0.92, Max. = 3.00).

### 2.3. Procedure

First, interested mothers received information about the study procedure and, in particular, about the procedure of BO sampling. After informed consent, they received their study kit containing all materials for sampling the BOs of mother and infant. Those were a t-shirt and a beanie for the mother, as well as a onesie and a beanie for the infant (all made of 100% cotton). Beforehand, the clothing was washed with odourless detergent (Denk mit Vollwaschmittel Ultra-Sensitive, dm-drogerie markt GmbH & Co. KG, Karlsruhe, Germany, www.dm.de, accessed 18 June 2021), stored in a re-sealable plastic bag, and labelled. The participants received the same odourless detergent to wash their bed linen and all the clothing that was additionally worn during the sampling night. Additionally, the participants were given an odourless medical shower gel (EUBOS flüssig wasch + dusch, Dr. Hobein GmbH, Meckenheim, Germany, www.eubos.de, accessed 18 June 2021) with which to wash themselves (body and hair) and their infants the evening before the BO sampling. They were instructed to refrain from wearing perfumed care products during the test procedure. The study kit further included a printed questionnaire about the sleeping situation for both mother and infant, smoking, pets, duration of wearing the sampling clothes, and contamination of the samples (e.g., urine). Information on diet was not assessed. 

No participant had to be excluded due to odour contamination. We furthermore tested whether the sleeping situation (same bed/same room different bed/different room) of mother and infant did influence the evaluation of the own infant’s body odour or the infant kin-recognition performance. As this was not the case (compare Appendix B.1.1), this variable was not included in further examination.

Mothers and infants were asked to wear their t-shirt/ onesie and beanie for one night. The next morning, the mothers put each piece of clothing back in one re-sealable plastic bag each and brought it back to our laboratory. There, the axillary area from every t-shirt/ onesie was cut out, stored in a separate plastic bag. Next, the axillary samples, the remaining upper body area, and the beanies were frozen at −25 °C until the experiment. 

#### Experimental Sessions: BO Matching and Rating

As an initial task, participants assessed pleasantness and intensity of two children’s BO samples that were collected for another study. That functioned as an anchor for what intensity to expect from the samples due to the overall low intensity of BO samples. For the main experiment, subjects were presented with the smell of axilla, breast, and head odour samples of themselves, of their own child, and of two unfamiliar mother–child dyads. The unfamiliar dyads were matched according to the child’s gender so that every mother had to smell at least one boy and one girl. In addition, participants smelled one unworn control sample that was also washed with the same odourless detergent beforehand and was stored in a plastic bag. Three body sites of six subjects plus the blank sample resulted in 19 samples in total for each mother. All samples were defrosted 30 min prior to the experiment. Each BO sample was rated for pleasantness and intensity on a visual analogue scale ranging from 0 (“unpleasant”/“not intense”) to 100 (“very pleasant”/“very intense”). In addition, participants were asked to identify their own and their own infant’s odour. Identification was carried out for each body site respectively: For example, mothers were presented three infant axillary odour samples and asked which one belonged to their own infant. For each identification trial, one out of three samples belonged to their infant or to themselves respectively. 

The same procedure was repeated for all body sites and for both maternal and infantile BOs individually. BO presentation and randomization was conducted as follows: Infantile odours were to be assessed first, then identified, followed by maternal odours, and finally these had to be identified as well. Within these fixed categories, we randomized the order of the samples (Figure 1). During smelling, participants had to close their eyes to minimize distractions and to prevent them from seeing whether they were being presented a beanie or an armpit/breast odour sample. 

Additionally, they were told prior to the presentation of the samples that they should indicate whenever they needed a break. After the presentation of each sample, mothers had 5 s to smell and concentrate on the odour. Then they rated each sample right away, before the next one was presented. This resulted in approximately 15 s in between the evaluations. Furthermore, we incorporated short breaks between the trial types (rating/recognition task), in which we announced and explained the next task.

We asked the mothers which part of the body they liked the odour of their infant the most and which part of the body they liked it the least. The mothers could answer in a free text field. Later, these answers were categorised. Beyond that, mothers were asked if they were familiar with the body odour of their own infant (yes/no).

### 2.4. Statistical Methods

Data were analysed with IBM SPSS Statistics 27 (IBM Corp. Released 2020. IBM SPSS Statistics for Windows, Version 27.0. Armonk, NY: IBM Corp) and JASP 0.14.0.0 (JASP Team (2020). JASP (Version 0.14) [Computer software]).

#### 2.4.1. Perceived Quality (Pleasantness and Intensity Ratings) of the BO Samples

Body site: 

Infants: Body site (axilla, breast, head) affects pleasantness perception, with highest ratings for the head odour, while intensity ratings do not differ between the body sites. 

Mothers: Body site (axilla, breast, head) affects pleasantness perception, with lowest ratings for axilla odour and affects intensity perception with highest ratings for axilla odour. 

Familiarity:

Infants: The own infant’s BO is perceived as more pleasant and less intense than unfamiliar odours.

Mothers: The own BO is perceived as more pleasant and less intense than unfamiliar odours.

Two repeated measures ANOVAs were used to assess the effects of body sites, adding post hoc analyses. We calculated separate repeated measures ANOVAs for infantile and for maternal Bos, each with the dependent variables pleasantness and intensity. The familiarity factor had two levels (own, unfamiliar), and the body part factor had three levels (axilla, breast, head). Additionally, Bonferroni-corrected post hoc analyses were run.

Furthermore, a Bayesian repeated measures ANOVA was run in order to obtain precise information about the probability of the different models (H0 model: no differences in BO quality due to body site, H1 (alternative) model: differences in BO quality due to body site), which extends informative power in the limited sample size.

To examine whether lower intensity of BO samples led to greater uncertainty in hedonic ratings, we performed a comparison of variances with respect to pleasantness ratings. For this purpose, we divided the ratings of both infantile and maternal odours into a low-intensity and a high-intensity group on the basis of the medians and then conducted a Levene’s test for the comparison of the variance (compare Appendix B.1.2).

In addition, we conducted an exploratory investigation into the relationship between pleasantness and intensity ratings using Pearson correlation coefficients for both infants and mothers and for familiar and unfamiliar samples separately.

#### 2.4.2. Recognition of the BO Samples (Own Infant’s BO Sample, Own BO Sample)

Comparison of body sites:

Infants: Body site (axilla, breast, head) affects recognition performance, with highest recognition rate for the head odour. 

Mothers: Body site (axilla, breast, head) affects recognition performance, with highest recognition rate for the axilla odour.

Recognition is higher than would be expected by chance:

Infants: All body sites lead to recognition of the own infant’s odour above chance.

Mothers: All body sites lead to recognition of the own odour above chance.

Chi-square tests were used to compare the identification performance between body parts for both groups. An additional Bayesian contingency table analysis was carried out. Binomial tests were applied in order to compare the identification performance to the chance level of 1/3. Likewise, a Bayesian binomial test was conducted to assess the likeliness of each model for the results (H0 Model: no differences with regard to the body site; H1 Model: differences with regard to the body site).

## 3. Results

Descriptive analyses of pleasantness and intensity ratings are presented in Table 2. 

### 3.1. Perceived Quality (Pleasantness and Intensity Ratings) of the BO Samples

#### 3.1.1. Body Site

Infants: Body site significantly affected pleasantness ratings (*F*(2, 54) = 3.719, *p* = 0.031, *ƞ**^2^* = 0.121). Post hoc pairwise comparisons revealed a trend for the difference between breast and head (*p* = 0.057), with the head being perceived as less pleasant, while the other comparisons did not indicate further differences (see Figure 2). Bayesian analysis, however, did not confirm that the head smelled less pleasant than breast and axilla, as this analysis neither supported the null (= no difference between body sites) nor the alternative (difference between body sites) model (BF_10_ = 1048). Regarding the perceived intensity, body site did not affect maternal ratings (*F*(2, 54) = 0.236, *p* = 0.791, *ƞ**^2^* = 0.009, see Figure 2) and Bayesian analysis strongly supported that all body sites were perceived as equally intense (BF_10_ = 0.072, the null model).

Mothers: Body site did not influence pleasantness (*F*(1, 54) = 1.391, *p* = 0.258, *ƞ**^2^* = 0.049) but significantly affected intensity ratings of the maternal odours (*F*(2, 54) = 5.181, *p* = 0.009, *ƞ**^2^* = 0.356). Post hoc pairwise comparisons revealed lowest intensity perception of the breast odour when compared with axilla and head (axilla versus breast: *p* = 0.016; breast versus head: *p* = 0.016). Bayesian analysis confirmed the findings with moderate evidence (BF_10_ = 0.212, the null model) for pleasantness and with suggestive evidence for intensity ratings (BF_10_ = 2.161, in favour of the alternative model).

For the infantile BOs, variances of pleasantness did not differ between the low-intensity and the high-intensity groups (*p* = 0.250), indicating that low intensity did not lead to greater uncertainty of hedonic ratings. Regarding the maternal BOs, we found that the variances differed between the groups (*p* < 0.001) in that the low-intensity group provided the smaller variance, indicating that more intensely perceived BOs were associated with greater spread of hedonic ratings. For details, see Appendix B.1.2.

#### 3.1.2. Familiarity

Infants: Perceived pleasantness (*F*(1, 27) = 2.126, *p* = 0.156, *ƞ**^2^* = 0.073) did not differ between the evaluation of one’s own and an unfamiliar infant’s odour, although descriptive data indicated a higher mean value of the pleasantness ratings for one’s own infant (across all body sites, own infants: *M* = 60.08, *SD* = 19.04, unfamiliar infants: *M* = 56.44, *SD* = 15.57). This was supported by the Bayesian analysis (BF_10_ = 0.490, suggestive evidence in favour of the null model). Familiarity significantly affected intensity ratings (*F*(1, 27) = 10.48, *p* = 0.003, *ƞ**^2^* = 0.280), indicating that unfamiliar odours were perceived as more intense compared with those of the own child, which was confirmed by Bayesian analysis (BF_10_ = 30.366, strong evidence in favour of the alternative model).

Mothers: Familiarity did not affect pleasantness (*F*(1, 27) = 2.301, *p* = 0.141, *ƞ**^2^* = 0.079) but did affect intensity ratings of the maternal odours. Mothers perceived their own BO as less intense (*F*(1, 27) = 11.378, *p* = 0.002, *ƞ**^2^* = 0.296). Bayesian analysis supported these results for pleasantness (BF_10_ = 0.528, suggestive evidence in favour of the null model), and for intensity ratings (BF_10_ = 68.886, strong evidence in favour of the alternative model). See Figure 2 for visualisation of the influence of body site and familiarity on pleasantness and intensity perception.

#### 3.1.3. Exploratory Analyses

Infants: Pleasantness and intensity were positively associated (*r* = 0.389, *p* = 0.041) for evaluation of the own infant’s head, while a negative relation occurred for unfamiliar infants’ heads (*r* = −0.412, *p* = 0.029). For infantile axilla and breast, no significant relations were observed for unfamiliar or familiar odours.

Mothers: A negative correlation of pleasantness and intensity was observed for evaluation of the axilla of unfamiliar women (*r* = −0.539, *p* = 0.003), while there was no effect for their own axillary BO. For maternal breast and head, pleasantness and intensity ratings were not associated with either unfamiliar or familiar odours. For details, see Appendix B.1.3.

### 3.2. Recognition of the BO Samples (Own BO Sample, Own Infant’s BO Sample)

#### 3.2.1. Comparison of Body Sites

Infants: There were no differences between recognition performance with regard to the body site (χ^2^(2) = 1.28, *p* = 0.53, d = 0.438), and the Bayesian analysis supported this with moderate evidence (BF10 = 0.164, in favour of the null model).

Mothers: Likewise, recognition did not differ between body sites for maternal odours (*χ^2^*(2) = 0, *p* = 1.00, d = 0), which was confirmed by the Bayesian analysis (BF10 = 0.079, strong evidence in favour of the null model). See Figure 3 for visualization.

#### 3.2.2. Recognition Is Higher Than Would Be Expected by Chance

Infants: The odour of one’s own infant was recognised at the axilla by 67.86% (CI: 49–86%), at the breast by 57.14% (CI: 38–77%), and at the head by 53.57% (CI: 34–73%). Thus, the recognition performance of the familiar odour was significantly above chance level of 33% at all body sites (all pi < 0.02, g = 0.265). Bayesian analysis supported this result (evidence in favour of H1) but indicated that evidence was lowest for recognition of the head (BF₊₀ = 2.542), followed by the breast (BF₊₀ = 7.162), with strongest evidence for the axilla (BF₊₀ = 362.79).

Mothers: For maternal odours, the recognition rate resulted in 67.9% (CI: 49–86%) across all body sites. Similarly, at all body sites, the familiar odour was recognised well in excess of the chance level of 0.33 as decisively confirmed by the Bayesian binomial test (all pi < 0.001, g = 0.348; BF₊₀ = 362.79 at all body sites, in favour of the alternative model). For details, see Appendix B.1.4.

When asked which part of the body the mothers would like to smell their infant the best, 82.14% (23 subjects) stated the head, 7.14% (2 subjects) the feet, 7.14% (2 subjects) the neck and 3.57% (1 subject) the mouth. When asked the opposite question about where they would least like to smell, 32.14% (9 subjects) named the nappy area, 25.00% (7 subjects) the feet, 17.86% (5 subjects) “nowhere”, 14.28% (4 subjects) “behind the ears”, and 3.57% (1 subject) each said the head, neck, hands, or groin folds. A total of 92.9% (26 subjects) of the mothers stated that they would be aware of their infant’s BO.

## 4. Discussion

Regarding the infants’ BOs, the head odour was, contrary to our expectations, perceived as least pleasant by trend. In line with our hypothesis, body site did not influence intensity ratings, but familiarity affected the perceived intensity, as one’s own infant smelled less intense than unfamiliar infants. In addition, mothers were able to recognise the BO of their own infants at every site and did so equally well. Body site did not impact pleasantness perception of the maternal Bos, yet the breast was perceived as least intense. As with the infants, we showed that mothers evaluated their own BO as less intense than BOs of other women. According to our expectations, the mothers could recognise their own odour above chance, and performance did not differ between body sites.

This study aimed at exploring BO perception in relation to maternal and infantile odours sampled at different sites of the body. According to a questionnaire study, the head of young infants is a prominent source of pleasant, affective feelings [44]. In line with these findings, 23 (84.14%) mothers stated in the present study that they liked their infant’s odour the most at the head. However, this statement differed from the experimental results. In view of other experimental studies on children’s BOs, it is not surprising that the infantile axillary odour was perceived as very pleasant [8,43]. Hence, the findings presented here might point to differences between questionnaire and experimental studies. While questionnaire studies are, e.g., prone to memory bias [49], the mothers might underestimate familiarity of the axilla or breast odour of their child, and it has been unknown how those odour sources differ in their perceived quality. While the memory of the child on the arm and therefore the head as a prominent BO source is probably more accessible, the BO of axilla and breast may be similarly familiar through cuddling and changing or smelling the child’s clothes.

Another explanation might relate to prior findings, which indicate that the exposure of an odour affects its perception. As Kaitz and colleagues [25] showed, the duration of a mother’s exposure to her newborn’s BO influences how well she can recognise her new-born through olfactory stimuli. In this study, we did not directly test exposure length, as the infants were already several months old, and hence, the participating mothers were already exposed to their infant’s odour for this time period. However, as mentioned above, the head is assumed to be the most exposed body site of an infant to its mother. Therefore, it is possible that the head is most often mentioned as the site of an infant’s body that smells best because it is the most exposed and not because it actually smells better than some other sites of the body (e.g., axilla and breast).

For the hedonic perception of maternal BO, there was no preference for a specific body site. Nevertheless, our exploratory analyses delivered a trend for the axilla of unfamiliar women: the more intense the axillary odour, the less pleasant it was perceived. This finding is consistent with previous literature on the relation of malodour and axillary odour [35].

In addition, the different secretion in the different sites of the body is relevant—especially in adults, as many secretion processes that lead to the development of more severe body odours only come into effect during puberty [33]. In adults, the axilla is of great importance when it comes to implicit communication, e.g., via androstedadienone, a prominent substance found in the male axilla skin surface that increases a positive emotional state in women [50]. Additionally, the odour of the maternal breast contributes to implicit communication, e.g., between mother and child. Hence, the breast odour is assumed to be characteristic enough to be recognised, which could be caused, like at the axilla, by the presence of apocrine sweat glands [51] and Montgomery glands [32]. Regarding the head BO secretion, there is a particularly high density of sebaceous glands there [52]. According to Rajan and colleagues [53], the sebum secreted at the head contributes considerably to the individual’s BO, as it is produced rapidly so that BO is quickly regenerated—for example, after taking a bath. In order to find out how different degrees of secretion by sebaceous and other sweat glands are reflected exactly in different body sites, future studies require more precise examination. In particular, BOs should be evaluated according to more differentiated hedonic criteria, e.g., in a chemical expert panel or supplemented with a chemical extraction procedure in order to quantify objective odour compounds. This has been done for infant’s heads [54] but is still pending for other BO sources.

During testing, some mothers reported that their infants’ beanies fell off at night, which led to an inconsistent wearing duration of the beanies, so that no statement about the general sampling quality can be made. Those reports emerged during the ongoing data collection, so that we were not able to systematically track the sampling quality. Moreover, it is very difficult for mothers to precisely state the sampling quality, as the sampling took place during nighttime, and thus the infants were not constantly watched. To overcome such obstacles, sampling during daytime might be a solution. When a mother is awake, she can put the beanie back on her infant’s head more easily. Another option would be to sample the odour with a bandage wrapped around the head, which could stick a little better. In this way, the problem of the different head circumferences of the infants, which may have led to the beanies falling off, could also be solved. The above explained methodological issues may be the reason why there was no clear evidence in the Bayesian model for our hypothesis on the hedonic perception of infantile odours at different body sites. To test whether a greater rating uncertainty might have been due to lower intensity of BO samples, we compared variances of low- and high-intensity groups. This did not affect perception of infantile odours but did affect maternal odours, as pleasantness ratings showed a greater variance for intensely perceived BOs. This might have been due to more polarizing hedonic ratings in strong Bos, whether positive or negative, since odour perception shows huge interindividual difference, e.g., relating to genotypic variation [55]. The study presented here replicated previous findings on human BO perception. First, we found that mothers perceive infantile odours as generally pleasant [8] and that the odour of one’s own infant was rated as being less intense than unfamiliar odours [5]. Second, the same pattern was observed for the perception of familiar versus unfamiliar adult BOs. Third, according to prior studies, mothers were well able to identify their own infant and themselves by olfactory cues. Furthermore, we expanded previous findings on the olfactory-recognition abilities of one’s own kin [56] and of oneself [21] (by showing that this effect can be applied to different parts of the body.

We are aware of the limitations of the data presented here based on the small sample size. The resulting insufficient sensitivity of the sample could, e.g., have contributed to the lack of findings regarding a maternal preference for the BO of the own infant: While we investigated 28 mothers and observed only a tendency towards higher pleasantness ratings for one’s own child’s BO compared to unfamiliar BOs, prior studies reporting a strong preference effect included N = 50 mothers [43] and N = 167 mothers [8].

## 5. Conclusions

In conclusion, we demonstrated that it is possible to take informative BO samples from the axilla, breast, and head of both infants and adult women. This is substantiated in particular by the fact that the recognition of the familiar body odour worked well to the same extent for all sites of the body. To deepen the findings presented here, chemosensory panels and profiling of odours sampled from different body sites should be carried out in the future.

## Figures and Tables

**Figure 1 brainsci-11-00820-f001:**
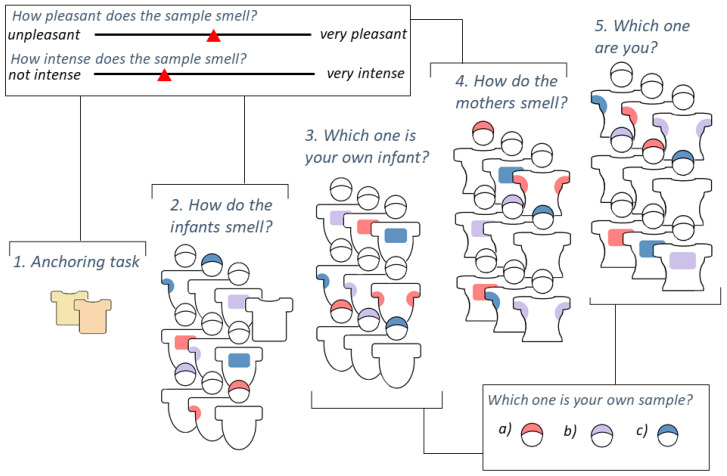
Visualisation of experimental procedure. Anchoring task (1.): smelling 2 unrelated child odours; (2.) Rating of infantile odours on three body sites (axilla, breast, head) plus blank control sample by pleasantness and intensity, randomised order of samples; (3.) Identification of own child for each body site, randomised order of body sites and samples; (4.) Rating of maternal odours on three body sites (axilla, breast, head) by pleasantness and intensity, randomised order of samples; (5.) Identification of own body odour for each body site, randomised order of body sites and samples.

**Figure 2 brainsci-11-00820-f002:**
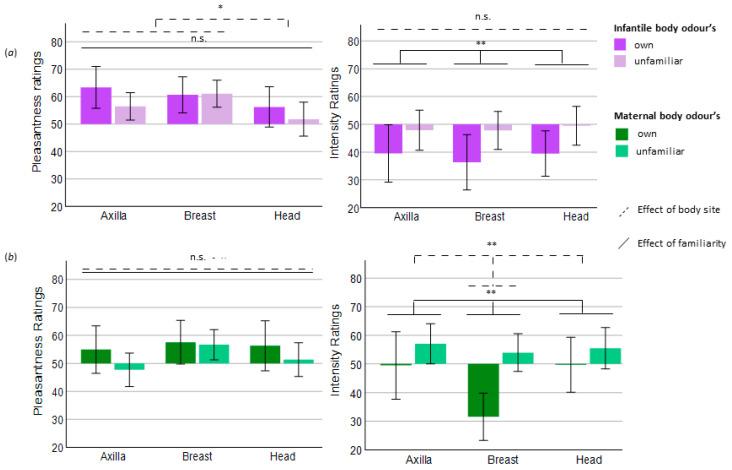
Perceived quality of the BO samples. (**a**) Infantile body odours. (**b**) Maternal body odours. * *p* < 0.05; ** *p* < 0.01; n.s. = not significant (*p* > 0.05). Pleasantness and intensity were measured with a visual analogue scale, ranging from 0 (“not pleasant”/“not intense”) to 100 (“very pleasant”/“very intense”). A rating of 50 therefore means a neutral evaluation of the odour, which is why this serves as the origin of the bars. Error bars: +/− 2 standard deviations.

**Figure 3 brainsci-11-00820-f003:**
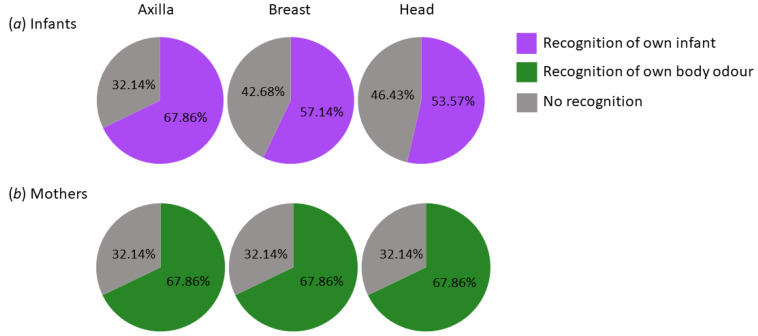
Recognition performance of the familiar odour. (**a**) Infantile odours. One’s own infant was to be chosen out of 3 body odour samples for each axilla, breast, and head of infants aged up to 12 months. (**b**) Maternal odours. Participants had to choose their own odour out of three samples from adult mothers for each axilla, breast, and head. Recognition of own infant/recognition of own body odour = the familiar odour was correctly identified. No recognition = participants mistook an unfamiliar odour for their own/their own infant’s one.

**Table 2 brainsci-11-00820-t002:** Descriptive analyses of pleasantness and intensity ratings. *n* = sample size, Min = minimum value, Max = maximum value, Mean = mean value, SD = standard deviation. All BOs were rated on a visual analogue scale ranging from 0 (“not pleasant”/“not intense”) to 100 (“very pleasant”/“very intense”).

		Infantile Axillae					Infantile Breast					Infantile Head				
		*n*	Min	Max	Mean	SD	*n*	Min	Max	Mean	SD	*n*	Min	Max	Mean	SD
**Pleasantness**	own	28	22	99	63.36	20.2	28	33	90	60.64	17.4	28	16	91	56.25	19.49
	unfamiliar	28	32	82.5	56.46	14.3	28	30	87	61.07	13.7	28	19	87	51.78	17.56
		**Maternal axillae**					**Maternal breast**					**Maternal head**				
		*n*	Min	Max	Mean	SD	*n*	Min	Max	Mean	SD	*n*	Min	Max	Mean	SD
**Pleasantness**	own	28	2	100	54.93	22.5	28	16	99	57.54	20.7	28	14	92	56.78	23.67
	unfamiliar	28	5	85	47.71	18.5	28	26	73.5	56.66	12.9	28	12	80.5	51.34	17.54
		**Infantile axillae**					**Infantile breast**					**Infantile head**				
		*n*	Min	Max	Mean	SD	*n*	Min	Max	Mean	SD	*n*	Min	Max	Mean	SD
**Intensity**	own	28	1	93	39.5	27.3	28	2	89	36.32	26.3	28	1	88	39.46	21.56
	unfamiliar	28	12	88	47.86	19.3	28	15	80.5	47.8	19.2	28	14	75	49.46	18.56
		**Maternal axillae**					**Maternal breast**					**Maternal head**				
		*n*	Min	Max	Mean	SD	*n*	Min	Max	Mean	SD	*n*	Min	Max	Mean	SD
**Intensity**	own	28	2	101	49.46	31.2	28	3	100	31.57	21.9	28	7	94	49.71	25.32
	unfamiliar	28	20	95.5	57.02	21.3	28	18.5	96.5	53.95	19.3	28	1.5	93.5	55.5	21.09

## Data Availability

The datasets generated for this study are available on request to the corresponding author.

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
