# Peer review of "Body Odours Sampled at Different Body Sites in Infants and Mothers—A Comparison of Olfactory Perception"

_brainsci, 2021, doi:10.3390/brainsci11060820_

Round 1

Reviewer 1 Report

In their study, the authors collected body odours of 28 mothers and their infants and compared three sampling locations: breast, axillae and head. They compared the intensity und pleasantness of the three sampling locations. Furtehr comparisons were between infants and adults (mothers) and whether the odour of the own body/infant could be recognised. In infants there was no significant effect of body site, neitehr for intenisty nor for pleasantness, for the mothers there was a main effect of body site only for intenisty: odours from the breast were rated as being less intense than odours from the axillae or the head. Own infants were rated as smelling less intense than unfamiliar infants, across all body sites. In line with previous research, mothers were able to recognise their own infants body odour.

This study is of importance for people conducting research on human body odours as it shows that the site from which the odour is sampled has only limited effect on the results. 

I have only a few comments which I would like to ask the authors to address before I can endorse publication:

1) The literature review (including Table 1) is slightly unbalanced, as most cited articles stem from the same research group. 

2) on Page 2 the authors list the procedures and behavioural restrictions used in various body odour studies. Probst et al. (2017) have included a concise description of things that should be taken into account (specifically see Probst et al.'s electronic supplementary material 1, and also Gildersleeve et al., 2012; or Lobmaier et al., 2018). Maybe the authors want to incorporate these studies in their review?

3) I understand that the authors asked about other people sleeping in the same room/bed and smoking etc. How was sharing the bed with a partner controlled for? Were these odours excluded? Or if the bed was shared with the infant? What where the specific behavioural and dietary restrictions and how were these enforced? If the authors only asked about other people sleeping in the same room 8and did nothing with this infomration), this may be a critical flaw and must at least be mentioned.

3) it seems that each participant sniffed 38(!) odours in one session (19 individual odour samples x 2). This seems rather a lot and I was wondering whether any breaks were included, and if yes, how many and how long?

Mini-issues:

-Some text seems to be missing after line 340 (around Figure 2) 

-Table 2: (formating problem) try to make "unfamiliar" fit on one line. 

- Reference 40: The format of this reference is different than the others (Laura Schaefer, M. Michael...)

Reviewer 2 Report

The manuscript (brainsci-1249959), title “Body odours sampled at different body sites in infants and mothers – a comparison of olfactory perception”, evaluates a very interesting topic in olfactory function. The study investigated whether different body odor sites, such as the axilla, breast and head, affect perceived body odor quality and recognition in both infants and adult women. The Manuscript is well written, newsworthy and clear to understand, but requires some minor revisions.

Specific comments:

In lines 54, 92 and 139 Authors should use lowercase character in the text.

Lines 187-189 Authors should indicate a reference as regards screening tests corrected identification.

Line 403 delete double full stops at the end of the sentence.

Line 463 Authors should delete double full stops.

Lines 529 and 535 It should be "Pearson's correlation coefficient"
